# Molecular Interaction Mechanism between Aromatic Oil and High-Content Waste-Rubber-Modified Asphalt



**Yuan Yan [1], Xinxing Zhou [2], Ruiqie Jiang [1], Maoping Ran [1] and Xinglin Zhou [1,\*]**

[1] School of Urban Construction, Wuhan University of Science and Technology, Wuhan 430081, China; yanyuan@wust.edu.cn (Y.Y.); jiangruiqie@wust.edu.cn (R.J.); ranmaoping@wust.edu.cn (M.R.)

[2] Institute of Resources and Environmental Engineering, Shanxi University, Taiyuan 030031, China; zxx09432338@whut.edu.cn

\* Correspondence: zhouxinglin@wust.edu.cn

**Abstract:** High-content waste-rubber-modified asphalt (HRMA) has high viscosity and poor storage stability. HRMA not only improves the properties of road asphalt, but also reduces the environmental pollution caused by waste tires. Enhancing the molecular interaction of waste rubber and asphalt is key to making full use of HRMA. In this paper, aromatic oil was used as the activator for waste rubber. The molecular interaction mechanism between aromatic oil and HRMA was investigated. The radial distribution function, diffusion coefficient, free volume, solubility parameter, and shear viscosity were calculated through molecular simulations. Storage stability, micromorphology, and adhesive force were measured via experiments. The adhesive force of HRMA−1 (4.9 nN) was lower than that of RMA (6.2 nN) and HRMA−2 (5.8 nN). The results show that aromatic oil can promote the dispersion of waste rubber, making the storage of asphalt systems stable. There exists a strong electrostatic force between rubber and asphaltenes and an intermolecular force between rubber and aromatic oil or aromatics, which makes the aromatic oil and aromatics of parcel rubber molecules and waste rubber highly soluble in asphalt. Molecular simulations confirmed the molecular interaction between rubber and aromatic oil, and aromatic oil was shown to reduce the viscosity of HRMA.

**Keywords:** rubber-modified asphalt; molecular interaction mechanism; aromatic oil; molecular simulation

## 1. Introduction

The regeneration of waste rubber has long been a major issue that needs to be solved in many countries around the world [1–3]. The random stacking and disposal of waste rubber causes serious environmental pollution and wasting of resources [4]. Waste rubber acts as a modifier to enhance the mechanical properties of asphalt, which is a good application for waste tires [5]. Waste-rubber-modified asphalt (RMA), a mixture of 20% waste rubber and base asphalt, first appeared in a British patent in 1843 [6]. However, the compatibility problem between waste rubber and asphalt not only restricts the engineering application of RMA, but also hinders the recycling of waste rubber. Owing to the rich aromatics in aromatic oil, aromatic-oil-activated waste rubber has secured its popularity in asphalt materials due to its environmental, economic, and additive advantages, especially in high-content waste-rubber-modified asphalt (HRMA) [7].

There are many kinds of aromatic oil that can activate RMA. For example, waste engine oil, as an additive, modified asphalt, shows great potential in reducing viscosity and enhancing compatibility [8]. On the other hand, the addition of waste engine oil damages the rutting and aging resistance of asphalt [9]. So, aromatic oil can significantly increase compatibility, while damaging the rutting. Waste mineral oil can improve the compatibility and flexibility of RMA and has a positive effect on the workability of RMA [10,11]. Waste cooking oil can obviously reduce the viscosity of RMA and improve its compatibility, low temperature performance and antiaging performance [12]. Waste cooking oil residue can significantly increase the compatibility of RMA. The improved compatibility of RMA

incorporating ground tire rubber pre-swelled with waste cooking oil residue was mainly attributed to the extended release of nature rubber and traces of synthetic rubber and inorganic filler into the asphalt phase [13]. Incorporating bio-oil-activated rubber is also an effective way to improve the compatibility of waste rubber and asphalt [10]. Bio-oil also reduces the viscosity of RMA and improves its workability [14]. Segregation was alleviated due to the microwave pretreatment of bio-oil on crumb rubber, which facilitated the swelling and depolymerization of crumb rubber in melt asphalt [15]. Grafting bio-oil onto rubber particles can increase the crossover modulus after thermal aging and cohesion to alleviate the moisture damage of RMA [16]. Pyrolysis bio-oil can improve this property through the rutting and fatigue parameters of RMA. Moreover, pyrolysis bio-oil can also reduce the viscous property and increase the elastic property of asphalt binder [17]. Bio-oil grafted onto a rubber surface via a heat treatment to create surface-activated rubber and bio-oil-activated RMA can improve the storage stability of rubber-modified asphalt. Bio-oil was successfully absorbed by rubber particles, significantly improving the dispersion of rubber in asphalt. Both grafting and pre-swelling of rubber particles using bio-oil improved the properties of RMA [18]. Bio-oil can deoxidize and devulcanize waste rubber. Moreover, grafting bio-oil onto waste rubber decreases the glass transition temperature of the asphalt system, thereby improving the low-temperature properties [19]. In summary, aromatic oil can reduce the viscosity and improve the storage stability of RMA. The viscosity and molecular interaction of HRMA are stronger than those of RMA, and it has a high waste rubber content. So, the relationship between viscosity and molecular interaction in HRMA is particularly challenging to elucidate compared to that in RMA. Moreover, the molecular interaction mechanism between aromatic oil and HRMA is not clear. The novelty of this work is that molecular dynamics simulations were used to evaluate the molecular interaction mechanism between aromatic oil and HRMA, and experimental methods were used to verify the simulated results.

In this study, aromatic oil was used to activate waste rubber, which was then used to modify asphalt. The radial distribution function, diffusion coefficient, free volume, solubility parameter, and shear viscosity were calculated through molecular simulations. Storage stability, micromorphology, and adhesive force were measured via experiments. The molecular interaction mechanism between aromatic oil and HRMA was positively analyzed.

## 2. Materials and Methods

### 2.1. Raw Materials

70# asphalt was bought from Hubei Guochuang Hi-tech Materials Co., Ltd (Wuhan, Hubei Province, China). Aromatic oil was bought from Hengshui Shengkang Chemical Co., Ltd. (Hengshui, Hebei Province, China). Waste rubber (40 mesh) came from Changzhou Rong Ao Chemical new materials Co., Ltd (Changzhou, Jiangshu Province, China). HRMA with 35% waste rubber content was prepared. The 70# base asphalt was heated to a flow state at 130 °C~135 °C and poured into the stainless-steel sample holder. Aromatic oil accounting for 20% or 35% of the waste rubber was added with microwave irradiation at 1000 W (irradiation power) and 60 s (irradiation time). Then, the aromatic-oil-activated waste rubber was added into asphalt and heated to 170 °C~180 °C. Finally, a high-speed shear meter was used for continuous shearing for 60 min at a rate of 2000 r/min and a temperature of 170 °C~180 °C to obtain the HRMA sample. The fundamental properties of the raw materials are shown in Table 1.

**Table 1.** The fundamental properties of raw materials.

| Material | Penetration (25 °C, 0.1 mm) | Penetration Index (PI) | Softening Point (°C) | Ductility (5 °C, cm) |
|---|---|---|---|---|
| Asphalt | 64 | 0.90 | 49 | 21 |
| HRMA | 52 | 0.85 | 62 | 35 |
| | Density (g/cm$^3$) | Flash point (°C) | Kinematic viscosity (mm$^2$/s) | Condensation point (°C) |
| Aromatic oil | 1.02 | 218 | 17 | −5 |
| | Specific gravity | Moisture content (%) | Sieving rate (40 mesh, %) | Ash content (%) |
| Waste rubber | 1.13 | <1.0 | ≥95 | ≤1.0 |

### 2.2. Molecular Simulations of HRMA

#### 2.2.1. Molecule Building

As shown in Figure 1, for asphalt, we selected asphaltenes, resins, aromatics, and saturates (SARAs) four-component models, and the SARA components were composed of molecules previously elucidated by Li [20]. Aromatic oil is represented by xylene ($C_8H_{10}$). Waste rubber is represented by styrene-butadiene rubber (SBR, $C_{88}H_{86}$). Waste-rubber-modified asphalt, abbreviated as RMA, included 20% waste rubber. The detailed molecular components of HRMA included 35% waste rubber and 100% asphalt, and it is abbreviated as HRMA−1. In order to improve the storage stability, 7% aromatic oil was added into HRMA, which is abbreviated as HRMA−2. The dosages of waste rubber and aromatic oil used were the same as those in the authors' previous study [21]. The molecular models were built using Materials Studio 7.0 software.

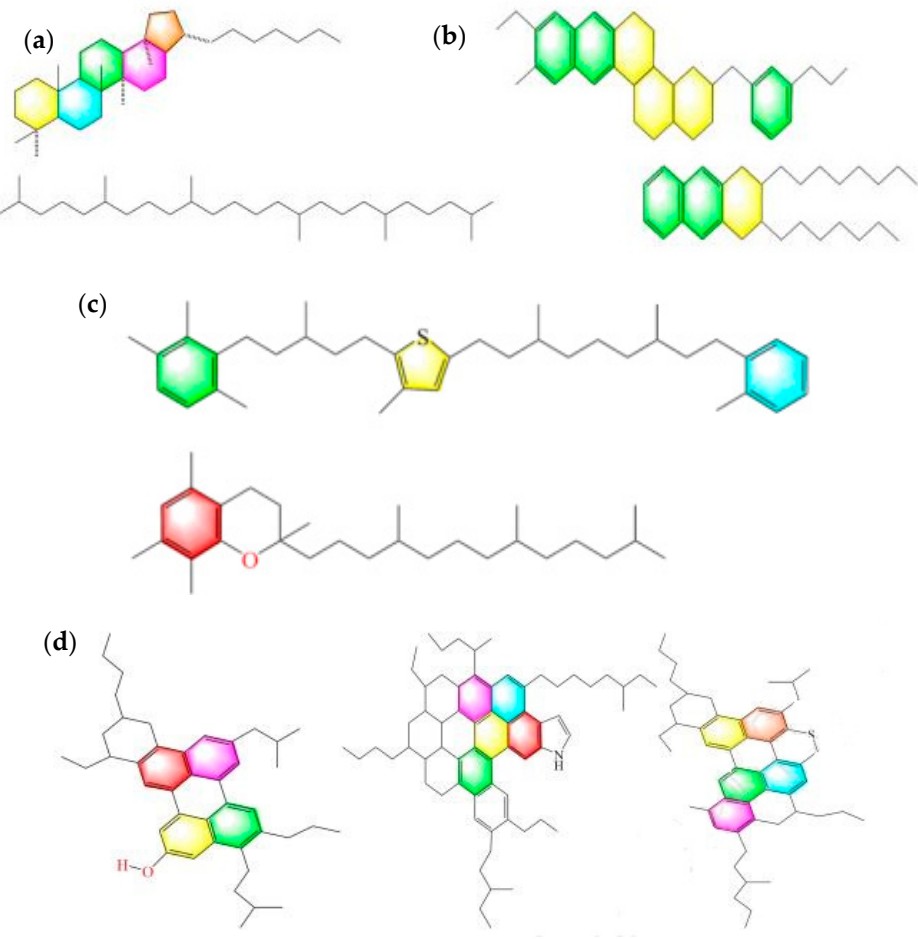

**Figure 1.** *Cont.*

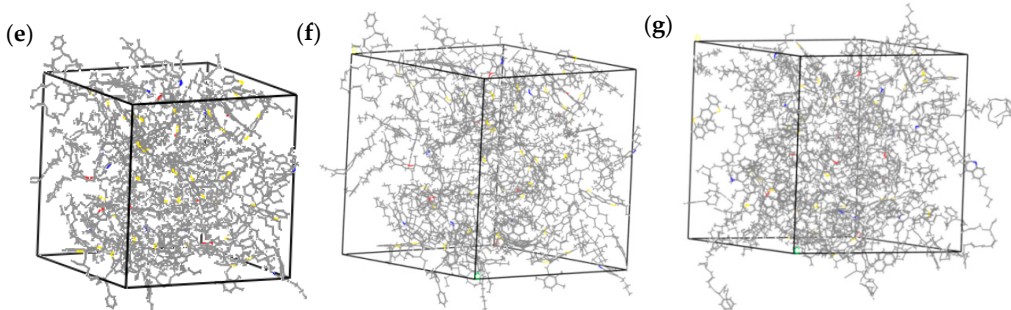

**Figure 1.** The four-component models of asphalt. Different ring molecules are shown in different colors in (**a**–**d**): (**a**) saturates, (**b**) aromatics, (**c**) resins, (**d**) asphaltenes, (**e**) RMA, (**f**) HRMA−1, (**g**) and HRMA−2.

The numbers of molecules for all components in the molecular models are shown in Table 2. Three rubber molecules were in RMA, and there were four rubber molecules in HRMA−1 and HRMA−2.

**Table 2.** The numbers of molecules for all components in the molecular models.

| Asphalt | Molecular Formula | Number of Molecules | Mass Fraction (%) |
|---|---|---|---|
| Resins | $C_{18}H_{10}S_2$ | 15 | 37.44 |
| | $C_{29}H_{50}O$ | 5 | |
| | $C_{66}H_{81}N$ | 2 | |
| Asphaltenes | $C_{51}H_{62}S$ | 3 | 34.39 |
| | $C_{42}H_{54}O$ | 3 | |
| Saturates | $C_{35}H_{62}$ | 4 | 14.36 |
| | $C_{30}H_{62}$ | 4 | |
| Aromatic | $C_{30}H_{46}$ | 13 | |
| | $C_{35}H_{44}$ | 11 | 13.81 |
| Rubber | $C_{88}H_{86}$ | 3/4 | |
| Aromatic oil | $C_8H_{10}$ | 22 | |

### 2.2.2. Simulation Details

An amorphous cell module was adopted to build the molecular model of RMA and HRMA by placing molecules into a cubic box through a construction task. The initial density was set to $1.0 \, \text{g/cm}^{-1}$ at room temperature (298 K) and pressure (1 atm) to conduct geometry optimization, which was used to obtain the energy minimization system. Then, dynamics simulation was applied for 100 ps, and the time step was set to 1 fs in the NPT ensemble using the Forcite module. Finally, the NVT ensemble was run again at 100 ps and the time step was set to 1 fs to equilibrate the system. In the molecular dynamics simulation process, we selected the universal force field, which was very suitable to predict the thermodynamic properties of the asphalt system.

### 2.2.3. The Calculation of Radial Distribution Function

The radial distribution function (RDF, $g(r)$), which is related to the orderliness and arrangement of molecule structures, can be calculated for either one or two sets of atoms. $g(r)$ represents the change law of the density function with distance from a reference position, which is the probability parameter of the occurrence of a particular molecule at a distance from the reference point $r$. The RDF is calculated using Equation (1).

$$g(r) = \frac{1}{\rho 4\pi r^2 \Delta r} \frac{\sum_{t=1}^{T} \sum_{j=1}^{N} \Delta N(r \rightarrow r + \Delta r)}{NT} \tag{1}$$

where $\rho$ is the density, $r$ is the distance in the radial direction, $\Delta r$ is the interval distance, $\Delta N$ is the interval number of atoms, and $N$ and $T$ are the total number of atoms and total time of the system, respectively.

### 2.2.4. The Calculation of Diffusion Coefficient

The diffusion coefficient ($D$) of molecules represents the movement capability of HRMA, which can be calculated through mean square displacement ($MSD$) using Equation (2). It is used to evaluate the fluidity and diffusion of the asphalt system and is related to molecular interaction. $MSD$ is a measure of the movement deviation of the molecular position from the initial position over time and is calculated with Equation (3).

$$D = \frac{1}{6T} MSD \tag{2}$$

$$MSD = |r(t) - r(0)|^2 \tag{3}$$

### 2.2.5. The Calculation of Free Volume

The free volume (FV) of molecules determines the possibility of molecular motion, and the greater the free volume, the greater the possibility of molecular motion. By calculating the Connolly surface, the FV and the occupied volume can be obtained.

### 2.2.6. The Calculation of Solubility Parameter

The solubility parameter ($\delta$) is an important parameter of polymers used to evaluate the interaction between waste rubber and asphalt. According to the Hansen solubility parameter theory, the $\delta$ of HRMA can be calculated using Equation (4).

$$\delta^2 = (\delta_D)^2 + (\delta_P)^2 + (\delta_H)^2 \tag{4}$$

where $D$, $P$, and $H$ represent the dispersion component, polar component, and hydrogen bond force, respectively.

### 2.2.7. The Calculation of Shear Viscosity

The shear viscosity ($\eta$) is a macroscopic expression of molecular mass. The viscosity of asphalt reflects the colloidal solubility of asphaltenes. The Rouse and Debye–Stokes–Einstein (DSE) theory can be used to study the viscosity of HRMA, which can be calculated using Equation (5). The calculation of viscosity is carried out according to the radius of gyration ($R_g$). The radius of gyration ($R_g$) denotes the distance between the hypothetical concentration point of a molecule with differential mass and the axis of rotation. $R_g$ can be calculated using Equation (6).

$$\eta = \frac{\rho R T R_g^2}{6MD} \tag{5}$$

$$R_g^2 = \frac{\sum m_i r_i^2}{\sum m_i} \tag{6}$$

where $\rho$ is the density, $M$ is the molecular mass, $R_g{}^2$ is the radius of gyration, $m_i$ is the mass of atoms, and $r_i$ is the radius.

### 2.2.8. The Calculation of Rubber Dispersion

The dispersion of rubber in the HRMA system was calculated through molecular dynamics simulations, and the dispersion morphology of rubber–asphaltenes, rubber–aromatic oil, and rubber–aromatics in the HRMA system was analyzed.

*2.3. Experimental Verification*

2.3.1. Storage Stability Test of HRMA

The storage stability of HRMA was evaluated via a segregation experiment and micromorphology. The segregation experiment of HRMA was conducted according to JTG E40-2014. The micromorphology of HRMA was observed under a fluorescence microscope (FM). The HRMA samples were heated in a translucent liquid, which was then coated on a slide, and finally, the FM samples were obtained.

2.3.2. Adhesive Force Test of HRMA

The adhesive force of HRMA was measured using atomic force microscopy (AFM, Brand: Bruker; Model: Dimension FastScan). The sample size was selected as 10 μm × 10 μm, the tip radius of the probe was 8 nm, and the tip height was 10 μm~15 μm. In the process of testing the tip of a probe, in which jump contact and the trip phenomenon occur and a force curve is formed, the lowest force should be subtracted from the highest point of the force curve to obtain the adhesive force. The test of bond strength was based on previous studies [19].

2.3.3. The Functional Group Test of HRMA

The functional group test of HRMA was performed with a Fourier infrared spectrometer (FT-IR). The test parameters included scanning time (32 times) and resolution ratio (4 cm$^{-1}$). Moreover, the tested wavenumbers ranged from 500 cm$^{-1}$ to 4000 cm$^{-1}$.

**3. Results and Discussion**

*3.1. Molecular Interaction Analysis of HRMA*

As shown in Figure 2a, the radial distribution function (RDF) results between rubber and other components of asphalt in the HRMA system show that the RDFs of different components were close, especially the RDFs between rubber and asphaltenes, because HRMA systems are typically amorphous structures. In this section, we exclude all other components of HRMA and consider only two kinds of molecules. The RDF value between rubber and asphaltenes was the highest, followed by the RDF value between rubber and aromatics, and finally the RDF value between rubber and saturates/resins. As shown in Figure 2b, the RDF results for aromatic oil and other components of rubber or the HRMA system show that the difference in RDFs between different components was significant due to the large amount of aromatic oil added. The RDF values between aromatic oil and aromatics were the highest. The results indicate that aromatic oil could change the orderliness and arrangement of molecules' structure in HRMA and contribute to dissolving rubber into asphalt. The reason for this is that aromatic oil envelops the rubber particles, creating a sol state similar to that of asphalt, which promotes movement within the asphalt. This conclusion is also consistent with a previous study [15].

The RDFs of asphaltenes in three kinds of HRMA systems are given in Figure 3. A previous study found that the four components of RMA had different first peak positions before and after adding waste rubber or aromatic oil [15]. The first peak positions of asphaltenes in RMA (20% waste rubber content), HRMA−1 (35% waste rubber content), and HRMA−2 (35% waste rubber content with 7% aromatic oil) appeared at 0.61 Å, 0.59 Å, and 0.62 Å, respectively. However, the first peak areas of asphaltenes in RMA, HRMA−1, and HRMA−2 were 3.25, 6.42, and 4.85, respectively. The addition of waste rubber or aromatic oil had a big effect on the aggregation of inner asphalt molecules because the main molecular structures of SARA are aromatics, hydrocarbons, and saturates, and they change significantly when waste rubber and aromatic oil are added. The intensity of the first peak represents the packing density of asphalt molecules and is related to the number of molecules [22]. The higher the intensity of the first peak, the steeper the structure and order degree of the four components, indicating that the addition of waste rubber or aromatic oil has a great influence on the aggregation and order degree of asphalt.

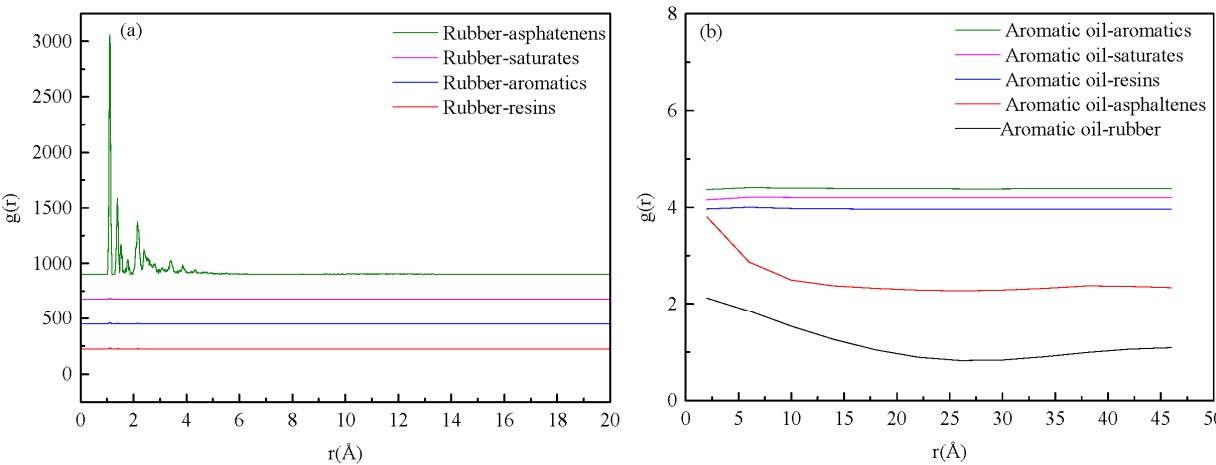

**Figure 2.** The radial distribution function of HRMA: (**a**) the *g*(*r*) of rubber and four components; (**b**) the *g*(*r*) of aromatic oil and components.

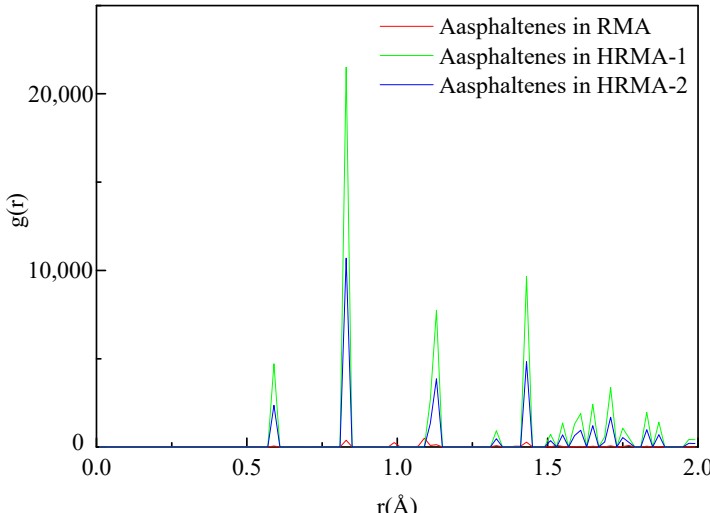

**Figure 3.** The radial distribution function of SARA.

Figure 4a depicts the mean square displacement (MSD) results for all asphalt systems. The MSD of HRMA−1 was lower than that of RMA, which indicates that the MSD of HRMA decreases with increasing waste rubber content. Moreover, The MSD of HRMA−1 was lower than that of HRMA−2, which indicates that aromatic oil can promote the diffusion of rubber. The MSD of different asphalt systems (RMA, HRMA−1, and HRMA−2) exhibits discrepancy with the addition of waste rubber or aromatic oil because rubber has a long molecular chain, which can inhibit the movement of asphalt molecules and reduce the MSD values. Aromatic oil has a short molecular chain, which can promote the asphalt molecules' movement and increase the MSD values. The results demonstrate that waste rubber can inhibit asphalt molecules' movement, while aromatic oil promotes asphalt molecules' movement and accelerates the dispersion of waste rubber in an asphalt system. This conclusion is also consistent with a previous study [21]. Figure 4b depicts the MSD results for asphaltenes in HRMA. The MSD of asphaltenes changed obviously with increasing simulation time. Furthermore, the MSD of asphaltenes decreased significantly with the increasing waste rubber content in the HRMA system, and the MSD of asphaltenes also decreased significantly after adding aromatic oil.

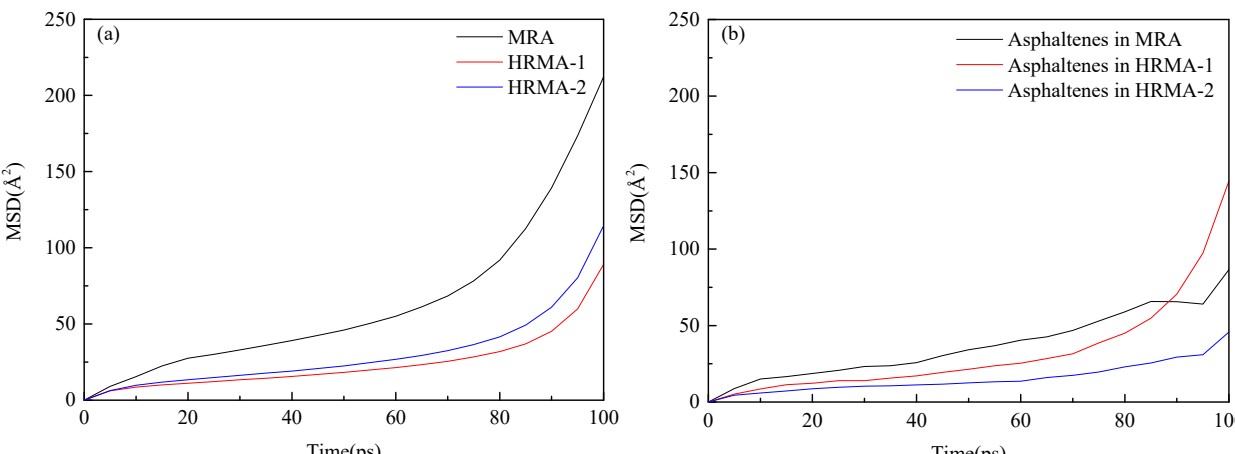

**Figure 4.** The mean square displacement of HRMA: (**a**) the MSD of asphalt systems and (**b**) the MSD of asphaltenes.

As shown in Table 3, the diffusion coefficient (D) of HRMA shows that the D of HRMA decreased with the increase in waste rubber content, while the D of HRMA increased with the addition of aromatic oil. In the HRMA system, the D value of aromatic oil was the largest, followed by aromatics, resins, saturated hydrocarbons, and finally asphaltene and rubber molecules. The results indicate that D is related to the length of the molecular chain, rubber molecules can inhibit the diffusion of asphalt molecules, and aromatic oil molecules can promote the diffusion of asphalt molecules. On the one hand, the relative molecular mass and the volume of waste rubber were also greater than the average of asphalt molecules, so rubber molecules make HRMA relatively more crowded and harder to move. On the other hand, aromatic oil molecules can dissolve some polar molecules in asphalt and promote the mobility of the asphalt system.

**Table 3.** The diffusion coefficients of HRMA.

| Asphalt | D ($10^{-7}$ cm$^2$/s) | Component | D ($10^{-7}$ cm$^2$/s) |
| :---: | :---: | :---: | :---: |
| RMA | 2.58 | Aromatic oil | 3.25 |
| HRMA$-$1 | 1.82 | Rubber | 1.12 |
| HRMA$-$2 | 2.45 | Aromatics | 2.89 |
| | | Resins | 2.52 |
| | | Saturates | 2.35 |
| | | Asphaltenes | 1.58 |

As described in Figure 5, as an example, the free volume (FV) of HRMA was obtained according to atom volumes and surfaces with different probes having a Connolly radius of 298 K. The free volume of MRA, HRMA$-$1, and HRMA$-$2 was 36,528.15 Å$^3$, 39,544.22 Å$^3$, and 45,224.57 Å$^3$, respectively. The results show that the FV of HRMA increased with the number of aromatic oil molecules, while the FV of HRMA decreased with the increase in rubber content. Furthermore, both rubber content and aromatic oil significantly affected the FV of HRMA. Usually, the greater the Connolly radius, the lower the FV of the HRMA system [23]. Therefore, the Connolly radius of HRMA will change with an increase in rubber content or the addition of aromatic oil. The FV of HRMA was greater than 5%, indicating that there is a critical FV that must be met to ensure that the molecules of an HRMA system can move freely.

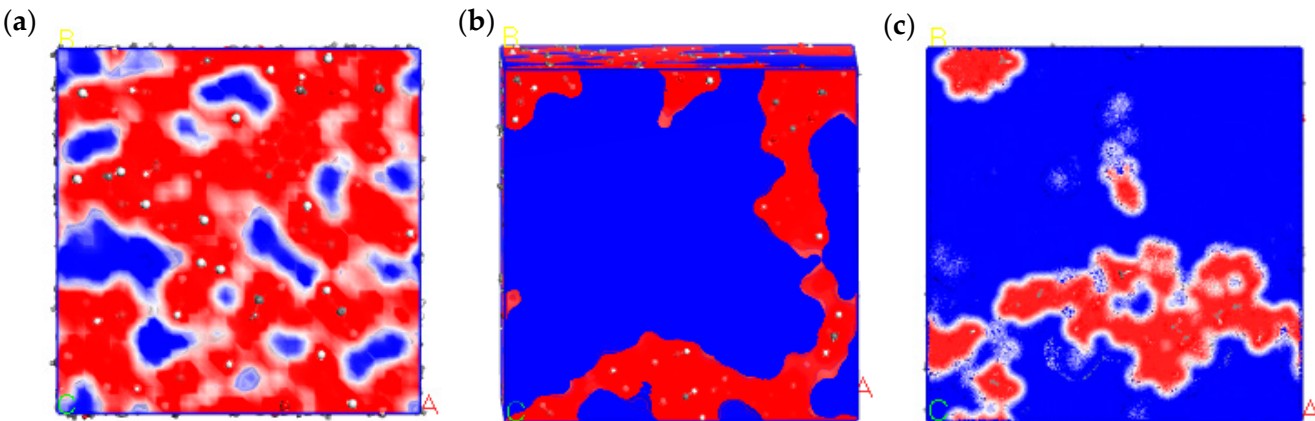

**Figure 5.** The free volume of HRMA: (**a**) RMA, (**b**) HRMA−1, (**c**) and HRMA−2.

Figure 6 shows the solubility parameter (δ) of HRMA before and after adding aromatic oil, obtained via molecular dynamics simulation. The δ of RMA, HRMA−1, and HRMA−2 was 12.5$(J/cm^3)^{0.5}$, 20.2$(J/cm^3)^{0.5}$, and 9.8$(J/cm^3)^{0.5}$, respectively. The results for δ indicate that the structure of HRMA significantly changed after adding aromatic oil, which weakened its compatibility. The addition of aromatic oil to HRMA is beneficial to reduce the difference in δ between rubber and asphalt systems to improve the compatibility of the components. This conclusion is consistent with a previous study [22]. However, an increase in waste rubber content would damage the compatibility of the components. The δ of different components exhibited differences in the HRMA system, and the δ of asphaltenes (16.7$(J/cm^3)^{0.5}$) and rubber (19.5$(J/cm^3)^{0.5}$) was higher than that of aromatic oil (5.2$(J/cm^3)^{0.5}$), aromatics (7.8$(J/cm^3)^{0.5}$), resins (15.1$(J/cm^3)^{0.5}$), and saturates (8.9$(J/cm^3)^{0.5}$). This indicates that the influence of different components on the δ of HRMA is significantly different.

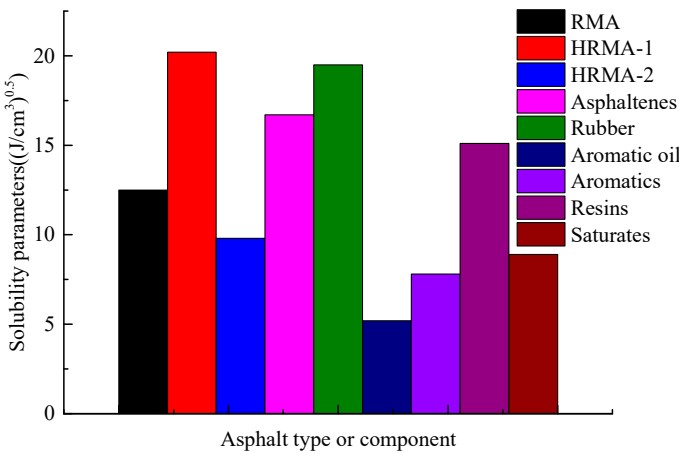

**Figure 6.** The solubility parameters of HRMA.

As shown in Figure 7, a segregation experiment was carried out to obtain the softening point difference. If the softening point difference is bigger than or equal to 2, it shows that the storage stability of HRMA is poor. If the softening point difference is less than 2, it shows that the storage stability of HRMA is good. The softening point difference of RMA (1.8 °C) was less than 2, which indicates that RMA has good storage stability and does not exhibit phase separation. The softening point difference of HRMA−1 (4.9 °C) was bigger than 2, which indicates that HRMA−1 has poor storage stability and exhibits phase separation. The softening point difference of HRMA−2 (1.5 °C) was less than 2, which indicates that HRMA−2 has good storage stability and does not exhibit phase separation.

The results show that with the increase in waste rubber content, the storage stability of HRMA becomes worse, and phase separation even occurs. The addition of aromatic oil improved the storage stability of HRMA. In other words, waste rubber will destroy the storage stability of HRMA, and aromatic oil will improve the storage stability of HRMA.

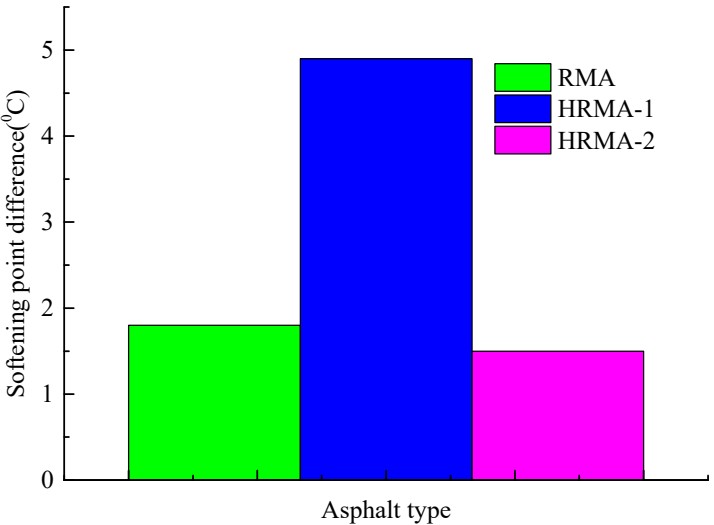

**Figure 7.** The softening point difference of HRMA.

The adhesive forces of HRMA were calculated using AFM data, and the adhesive force was equal to the stable value minus the minimum value. As shown in Figure 8, the adhesive force curve decreases at first, and then increases with the increase in the height sensor and the probe movement. The adhesive force of HRMA−1 (4.9 nN) was lower than that of RMA (6.2 nN) and HRMA−2 (5.8 nN), which indicates that the adhesive force of HRMA decreased with the increase in waste rubber content, while the adhesive force of HRMA increased with the addition of aromatic oil. This indicates that waste rubber can damage the adhesive properties of HRMA, and aromatic oil can improve the adhesive properties of HRMA. The reason for this is that waste rubber can form three-dimensional network structure compounds, which improve the adhesive properties of HRMA at a suitable content, while the excess waste rubber content can aggregate and damage the adhesive properties of HRMA. Aromatic oil can regulate four-component asphalt and improve the solubility of HRMA to increase the adhesive properties of HRMA.

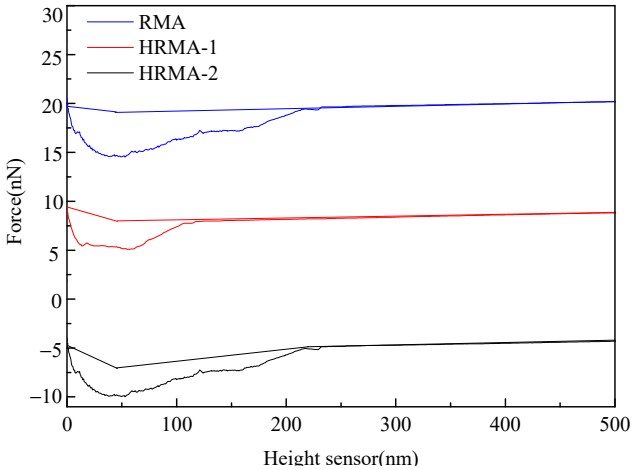

**Figure 8.** The adhesive force curves of HRMA.

The dispersion morphology of component molecules in HRMA is depicted in Figure 9. In order to determine the dispersion morphology of component molecules, the other components of HRMA were deleted, so only the selected molecules remained. The dispersion morphology of component molecules in HRMA showed that the rubber molecules were close to the asphaltenes and split by aromatic oil and aromatics. Moreover, there existed a strong electrostatic force between the rubber and asphaltenes and an intermolecular force between the rubber and aromatic oil or aromatics, which made the aromatic oil and aromatic parcel rubber molecules and the waste rubber highly soluble in asphalt. This indicates that aromatic oil can promote the dispersion of waste rubber, making the storage of the asphalt system stable.

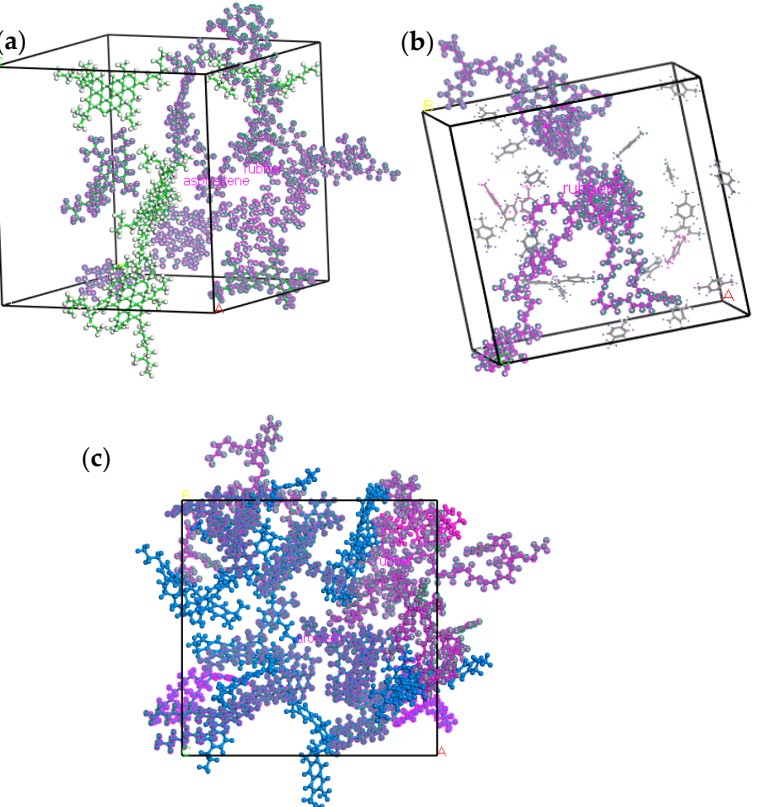

**Figure 9.** The dispersion morphology of component molecules in HRMA: (**a**) rubber–asphaltenes, (**b**) rubber–aromatic oil, and (**c**) rubber–aromatics.

### 3.2. Viscosity Reduction Mechanism Analysis of HRMA

As shown in Figure 10, the simulated viscosity of HRMA increased with the increase in waste rubber content, while the simulated viscosity of HRMA decreased with the addition of aromatic oil. The addition of aromatic oil is the key to viscosity reduction of HRMA. The experimental viscosity of HRMA shows a similar law of simulated viscosity. The Rouse and Debye–Stokes–Einstein (DSE) theory equation shows that reducing the viscosity of HRMA is needed to decrease the waste rubber content and aromatic oil must be added to reduce the radius of gyration and chain density. In addition, if the purpose of reducing the viscosity of HRMA is to be achieved, the molecular weight and diffusion coefficient of HRMA must be increased. The viscosity reduction mechanism of HRMA is as follows: the addition of aromatic oil promotes the movement of molecules and increases the diffusion coefficient, and the rotation radius of aromatic oil is smaller than that of waste rubber and asphalt components, thus reducing the simulated viscosity of HRMA.

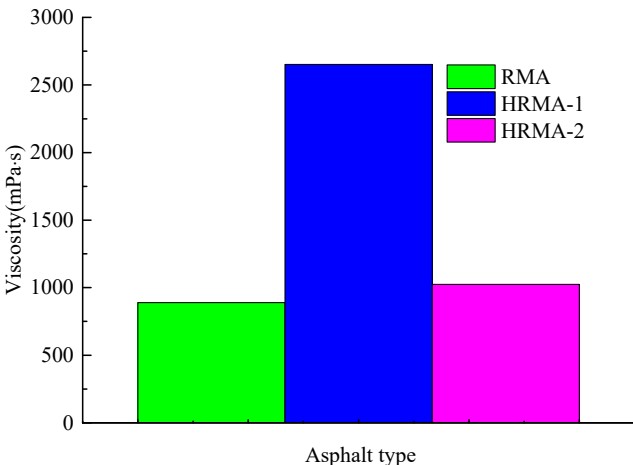

**Figure 10.** The simulated viscosity of HRMA.

As shown in Figure 11, the four-component asphalt results show that the addition of aromatic oil can increase the aromatic content of HRMA and decrease the asphaltene content to reduce the viscosity of HRMA. Moreover, aromatic oil can also be added to control the sol–gel structure of HRMA and change the stability of HRMA. However, the addition of aromatic oil had little effect on the content of resin and saturated matter. Previously, researchers found that the asphaltene content is related to the viscosity of asphalt, and the higher the asphaltene content, the higher the viscosity of asphalt [4]. The above results are in good agreement with previous conclusions.

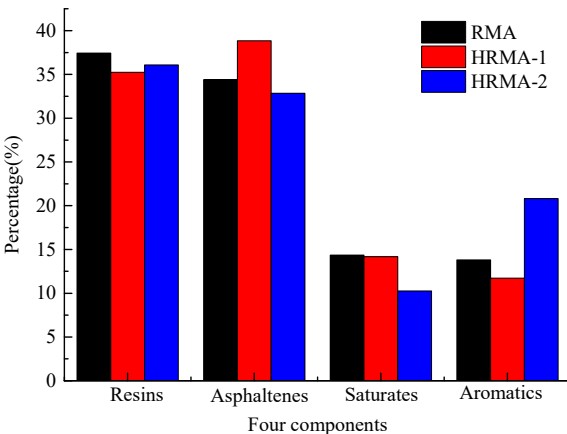

**Figure 11.** The four components of asphalt.

As shown in Figure 12, the FM morphology of HRMA shows that there were large quantities of black aggregate and yellow transparent phases in the HRMA system. Moreover, the yellow transparent phase areas increased and the black aggregate decreased with the addition of aromatic oil in the HRMA system. In combination with the properties of waste rubber, aromatic oil, and the four components, the black aggregate may be waste rubber aggregate and the yellow transparent phase may be the asphalt of aromatic oil or light components (aromatics and saturates). The results indicate that the viscosity of the yellow transparent phase was lower than that of the black aggregate. In addition, the viscosity reduction mechanism of HRMA was studied from a morphology perspective. It was found that the addition of aromatic oil increased the yellow transparent phase and reduced the black aggregate, thus changing the morphology of HRMA and reducing the viscosity of HRMA. The FM morphology of HRMA is related to the viscosity.

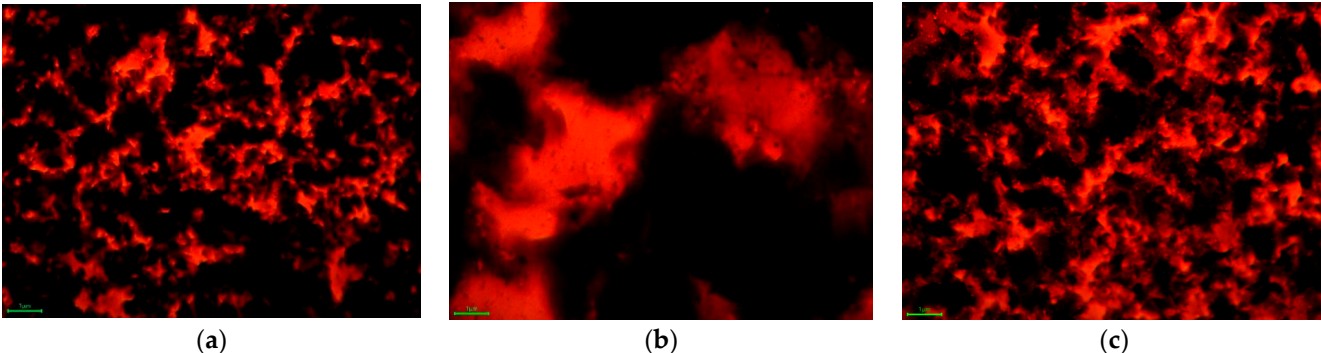

**Figure 12.** The fluorescence microscope morphology of HRMA: (**a**) RMA, (**b**) HRMA−1, and (**c**) HRMA−2.

As shown in Figure 13, the morphology of HRMA, as measured via AFM, shows that the four structure phases included the para-phase, peri-phase, catana-phase, and sal-phase. The catana-phase increased with the addition of aromatic oil in HRMA. Moreover, the peri-phase was the major structure phase in HRMA. Previous research found that the catana-phase was present in crystallization wax or asphaltenes [24]. The results show that aromatic oil may be present in crystallization wax. However, crystallization wax is a kind of low-viscosity aromatic oil. Therefore, the addition of aromatic oil can reduce the viscosity of HRMA by increasing the crystallization wax content.

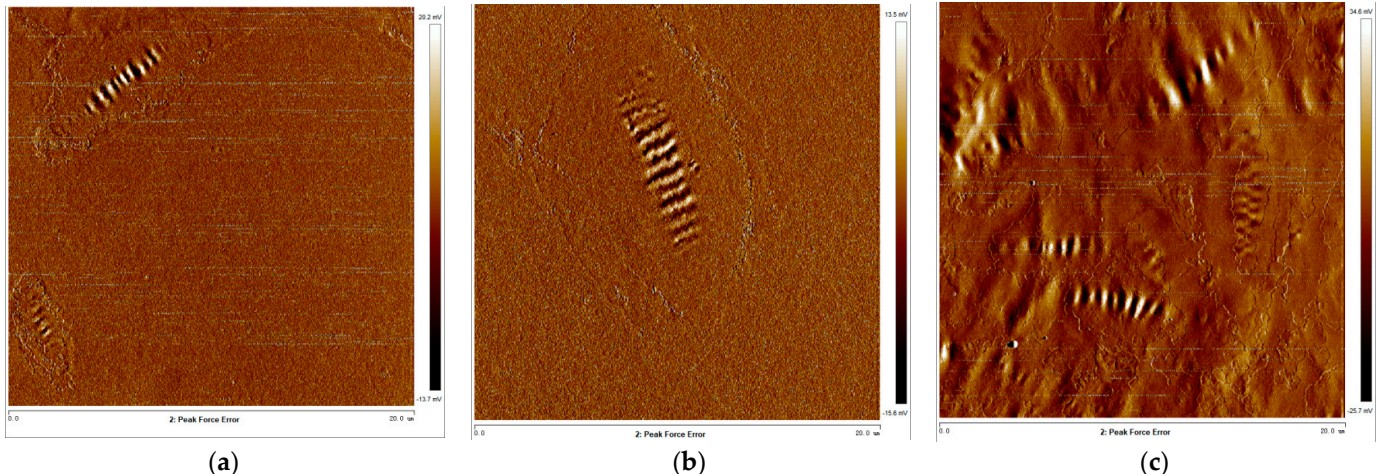

**Figure 13.** The atomic force microscopy morphology of HRMA: (**a**) RMA, (**b**) HRMA−1, and (**c**) HRMA−2.

As shown in Figure 14, the average surface roughness ($R_a$) of HRMA−1 and HRMA−2 was 22.5 nm and 8.59 nm, respectively. The average surface roughness of HRMA decreased with the addition of aromatic oil. This indicates that aromatic oil can change the microstructure of HRMA. The reason for this is that aromatic oil can decrease the average surface roughness of HRMA. The average surface roughness ($R_a$) of HRMA−1 is larger than that of RMA, which indicates that the average surface roughness of HRMA increases with increasing rubber content. The viscosity reduction mechanism of HRMA was analyzed from the perspective of atomic force microscopy. It was shown that the addition of aromatic oil could reduce the viscosity of HRMA by reducing the average surface roughness.

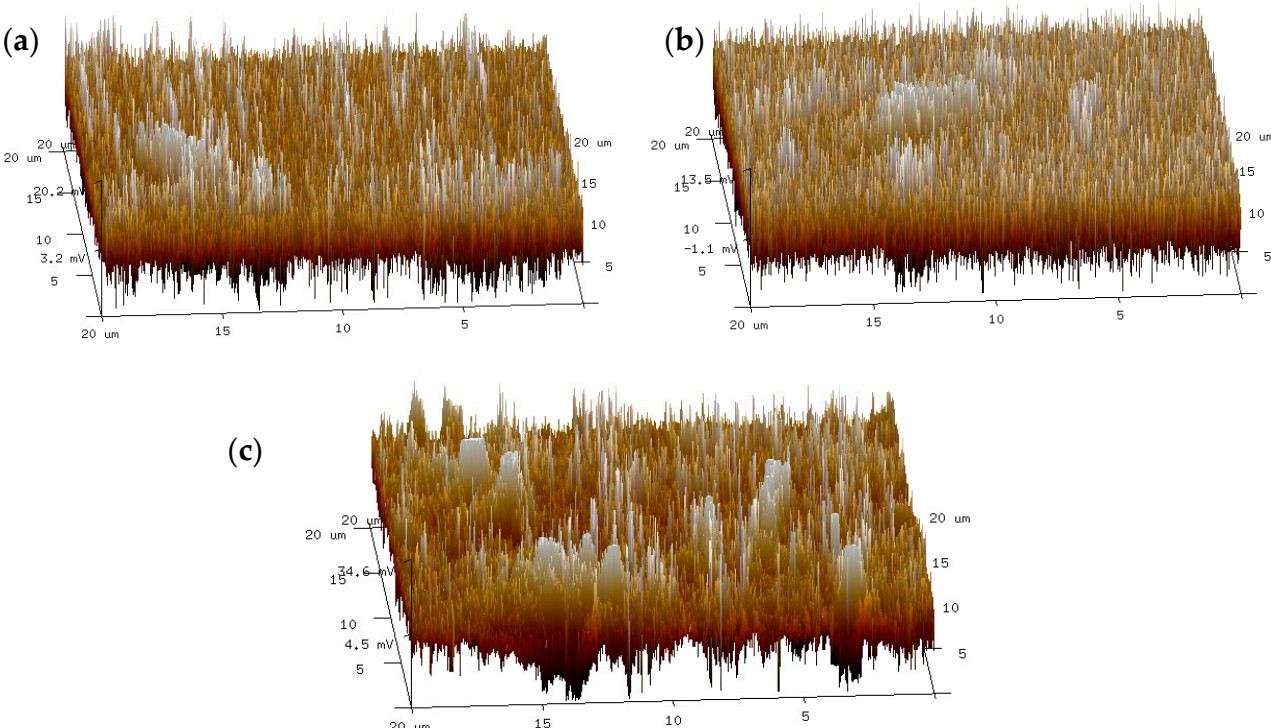

**Figure 14.** The average surface roughness of HRMA: (**a**) RMA, (**b**) HRMA−1, and (**c**) HRMA−2.

As shown in Figure 15, the FT-IR of HRMA shows that there were many characteristic peaks, which were mainly concentrated at 2918 cm$^{-1}$ (C−H), 2851 cm$^{-1}$ (C−H), 1460 cm$^{-1}$, and 1372 cm$^{-1}$. The greatest difference between the untreated HRMA and the HRMA with 7% aromatic oil added was the sulfoxide group band (S=O) near 1029 cm$^{-1}$. Moreover, there was a carbonyl peak (1720 cm$^{-1}$). The sulfoxide absorption peak gradually decreased and carbonyl peak increased after the addition of aromatic oil. These findings are in agreement with previous results [25]. Owing to the lower viscosity of HRMA with 7% aromatic oil compared to that of untreated HRMA, we concluded that the reduction in the sulfoxide absorption peak and the presence of a carbonyl peak reduce the viscosity of HRMA.

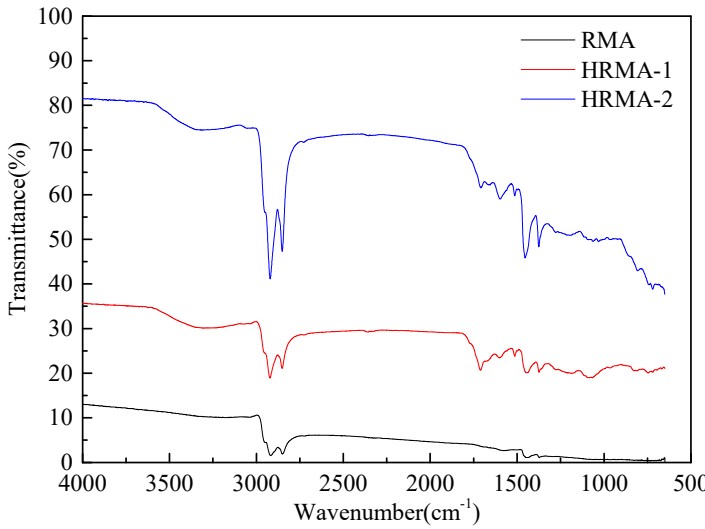

**Figure 15.** Fourier infrared spectrometer of HRMA.

## 4. Conclusions and Suggestions for Future Studies

The molecular interaction mechanism of HRMA was studied via molecular dynamics simulation and experiments. We can draw the following conclusions:

Aromatic oil can change the orderliness and arrangement of HRMA's molecular structure, which helps the rubber to dissolve into the asphalt. Furthermore, the addition of aromatic oil also has a great impact on the aggregation and orderliness of asphalt. The addition of aromatic oil can promote molecular motion and increase diffusion coefficients.

Aromatic oil can promote the dispersion of waste rubber and asphaltenes, making HRMA's storage more stable. In addition, molecular simulations confirmed the existence of molecular interactions between rubber and aromatic oil, which can reduce the viscosity of HRMA.

Despite the above conclusions, we are also aware of the limitations of this study, including the fact that the interaction mechanism between aromatic oil and rubber-modified asphalt was not studied in detail using the first principle, so our suggestions for future studies are as follows:

The interaction mechanism between aromatic oil and rubber-modified asphalt should be further studied using the first principle. In addition, the warm-mixing effect of aromatic-oil-activated rubber-modified asphalt must be further studied. Finally, an investigation of the activation effect of aromatic oil on waste rubber is also very necessary.

**Author Contributions:** Data curation, X.Z. (Xinxing Zhou); Formal analysis, X.Z. (Xinxing Zhou); Investigation, X.Z. (Xinxing Zhou) and R.J.; Methodology, Y.Y.; Resources, M.R.; Software, R.J.; Supervision, X.Z. (Xinglin Zhou); Validation, M.R.; Visualization, R.J. and M.R.; Writing—review and editing, Y.Y. and X.Z. (Xinglin Zhou). All authors have read and agreed to the published version of the manuscript.

**Funding:** The authors would like to acknowledge the financial support from the National Natural Science Foundation of China (NSFC: 51827812, 51778509, 52172392) and the Hubei Provincial Department of Science and Technology, the key research and development plan (2021BAA180).

**Institutional Review Board Statement:** Not applicable.

**Informed Consent Statement:** Not applicable.

**Data Availability Statement:** Not applicable.

**Acknowledgments:** The authors would like to acknowledge the financial and technical support from Wuhan University of Science and Technology (Wuhan, Hubei province, China) and Shanxi University (Taiyuan, Shanxi Province, China).

**Conflicts of Interest:** The authors declare no conflict of interest.

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
