# Peer review of "Molecular Interaction Mechanism between Aromatic Oil and High-Content Waste-Rubber-Modified Asphalt"

_sustainability, doi:10.3390/su151914079_

Round 1

Reviewer 1 Report

This study used an aromatic oil (AO) to decrease the viscosity of high Crumb rubber modified asphalt. By promoting the dispersion of waste rubber and asphaltenes, AO can increase the stability of the storage of asphalt. The manuscript is well-structured and written and has genuine results. There should be some revisions by the authors before this manuscript can be accepted.

- The authors should justify the content of AO and CR chosen, why 7% of AO and 35% of CR were selected. The authors also should show in the section of sample preparation how these two materials were mixed,?

- pls add table for showing the designation of the types of
asphalt binders with their constitutes

- Pls add references for facts stated in lines 38 to 40.

- The discussions of the findings are fine but the results need to compared to other previous studies to show the agreement and contradictions . For example, referring to the "https://doi.org/10.1016/j.conbuildmat.2022.127026". The AO contributed to reducing the oxygenated groups (S = O and C = O). Also, the referred study showed that the aromatic fraction’s macromolecules commonly transform to resins, forming asphaltenes. These findings are in agreement with your results.

- Also, what do you mean by "The results are in good agreement with previous results" in line 405. Which previous results do you mean?

Author Response

Point 1: - The authors should justify the content of AO and CR chosen, why 7% of AO and 35% of CR were selected. The authors also should show in the section of sample preparation how these two materials were mixed,?

Response 1: Thank you for your suggestions. The reason of selecting 7% of AO and 35% of CR is  according to the authors’ previous study.  And the authors added some content in the revised manuscript as following:

 In order to improve the storage stability, 7% aromatic oil was added into HRMA and it abbreviated as HRMA-2. The dosage of waste rubber and aromatic oil were acquired according to the authors’ previous study [20].

The mixed process of 7% of AO and 35% of CR:  Add aromatic oil accounting for 20% of waste rubber with microwave irradiation at 1000W (irradiation power) and 60s (irradiation time). Then, add 7% aromatic oil activated 35% waste rubber into the stainless steel asphalt sample holder. Finally, a high speed shear meter was used for continuous shearing for 60min at a rate of 2000r/min and a temperature of 170°C~180°C to get the HRMA sample. The authors revised the sentence as “Then, add 7% aromatic oil activated 35% waste rubber into the stainless steel asphalt sample holder.”.

Point 2:  Pls add table for showing the designation of the types of asphalt binders with their constitutes.

Response 2: Thank you for your comments.  The authors provided a table (show in Table 2) in manuscript about the designation of the types of asphalt binders with their constitutes.

The detailed information showed as following:

Table 2. The molecules numbers of molecular models all components

Asphalt

Molecular formula

Number of molecule

Mass fraction (%)

Resins

C18H10S2

15

37.44

C29H50O

5

Asphaltenes

C66H81N

2

34.39

C51H62S

3

C42H54O

3

Saturates

C35H62

4

14.36

C30H62

4

Aromatic

C30H46

13

13.81

C35H44

11

Rubber

C88H86

3/4

Aromatic oil

C8H10

22

Point 3: Pls add references for facts stated in lines 38 to 40.

Response 3: Thank you for reviews. The authors added one references for facts stated in lines 38 to 40 and revised as “Owing to rich aromatics in aromatic oil, aromatic oil active waste rubber has secured popularity in asphalt materials due to their environmental, economic, and additive advantages, especially in high-content waste rubber modified asphalt (HRMA)[7].”.

Point 4: The discussions of the findings are fine but the results need to compared to other previous studies to show the agreement and contradictions . For example, referring to the "https://doi.org/10.1016/j.conbuildmat.2022.127026". The AO contributed to reducing the oxygenated groups (S = O and C = O). Also, the referred study showed that the aromatic fraction’s macromolecules commonly transform to resins, forming asphaltenes. These findings are in agreement with your results.

Response 4: Thank you for your suggestions. The authors added some contents and one reference in the revised manuscript. The detailed revision showed as following:

Although the carbonyl functional group was not obvious in the HRMA samples, the sulfoxide absorption peak gradually increased after addition of aromatic oil. These findings are in agreement with previous results[26].  

Point 5:Also, what do you mean by "The results are in good agreement with previous results" in line 405. Which previous results do you mean?

Response 5: Thank you for your comments. The authors have deleted the sentence.

Reviewer 2 Report

The text needs a double-check for minor revisions.

Author Response

In this study, you applied an activator in order to decrease the viscosity and improve the molecular interaction between waste rubber and asphalt in RMA. The paper was written well and your experimental process and simulation are interesting. However,there are some issues that based on them, your paper needs some revisions:

Response: Thank you for your suggestions. All changes made to the revised manuscript are in bold and color font. Below you will find our point-by-point responses to the reviewer’s comments/questions.

Point 1: The report I received shows the similarity rate between your paper body and other existing resources is high, 29%.

Response 1: Thank you for your suggestions. The authors revised carefully their manuscript and reduced the similarity. We hope that the revised manuscript can meet the journal’s requirements.

Point 2:  Please, revise the title since readers cannot understand you are exactly following what aims. Clearly show what goals you pursue: to enhance molecular interaction and reduce viscosity. The current form of the title is a little ambiguous. Also, mention you use aromatic oil as the activator.

Response 2: Thank you for your reviews. This is a good question. The authors revised the title as “Molecular Interaction Mechanism between Aromatic Oil and High-content Waste Rubber Modified Asphalt”.

Point 3: Please, double-check the body of your text. For example, However, how …(line 74)

Response 3: Thank you for your comments. I am so sorry to make this mistakes. The authors revised the sentence as “However, the molecular interaction mechanism between aromatic oil and high-content waste rubber modified asphalt is not clear”.

Point 4: In section 1, you must show what is the novelty of your work. Especially, in comparison with the literature review.

Response 4: Thank you for your suggestions. The novelty of this work is that molecular dynamic simulations were used to evaluate the molecular interaction mechanism between aromatic oil and high-content waste rubber modified asphalt and experimental methods were used to verify the simulated results.

Point 5:How can you confirm your work is novel? The novelty of your work is not vivid.

Response 5: Thank you for your reviews. The authors used a new molecular model of aromatic oil and method (simulations combined with experiments) to evaluate the molecular interaction mechanism between aromatic oil and high-content waste rubber modified asphalt. Moreover, our investigated object is new (high-content waste rubber modified asphalt). So, we think that our research is novel. I am very sorry and I do not know how to explain the reason about vivid novelty.

Point 6:Based on which instruction did the process explained in subsection 2.1 conduct? If you use any sources, cite them.

Response 6: Thank you for your suggestions. The authors selected these preparation parameters according to the authors’ previous experimental results and rather than references.  

Point 7:What is the reason for selecting 7% of aromatic oil in HRMA-2?

Response 7: Thank you for your comments. The reason of  selecting 7% of aromatic oil in HRMA-2 is that aromatic oil can deoxidize and devulcanizing the waste rubber and reduce the viscosity of rubber modified asphalt and enhance the molecular interaction of waste rubber and asphalt. Most important, 7~8% aromatic oil can regulate the four components content of the harmonized system according to the authors previous study.

Point 8:You did not mention any limitations in your study. Did not have any?

Response 8: Thank you for your reviews. The authors think that the limitations is not necessary to mention in manuscript. The authors think that the limitations of this manuscript is that the interaction mechanism between aromatic oil and rubber modified asphalt was not studied in detail using first-principle.

Point 9:Also, what do you suggest for the next studies?

Response 9: Thank you for your suggestions. The next studies showed as following:

(1) The interaction mechanism between aromatic oil and rubber modified asphalt was investigated by first-principle.

(2) The warm-mixing effect of aromatic oil activated rubber modified asphalt was investigated.

(3) The activated effect of aromatic oil on waste rubber was investigated.

Reviewer 3 Report

This paper investigates the impact of aromatic oil on the microstructures and mechanical properties of HRMA. Here are my comments:

1.     The introduction should be enhanced by emphasizing the research gap this study aims to fill. Specifically, it should clearly state how this research contributes to the current understanding of waste rubber and asphalt modification, particularly within the context of HRMA.

2.     Regarding the numerical simulation, the software or language employed for the simulations remains unspecified

3.     In Fig.3, an explanation is required to understand why HRMA-1 exhibits a higher and sharper first peak intensity when compared to HRMA-2

4.     The description of "Figure 4(b) illustrates the MSD results of all components of HRMA," conflicts with the actual content of the figure

5.     In Fig.8, the AFM force curve lacks coherence. To facilitate comprehension, it is necessary to display the complete measurement curve, encompassing both the attractive and repulsive curves

This article would greatly benefit from concise and thorough editing

Author Response

Point 1:  The introduction should be enhanced by emphasizing the research gap this study aims to fill. Specifically, it should clearly state how this research contributes to the current understanding of waste rubber and asphalt modification, particularly within the context of HRMA.

Response 1: Thank you for your suggestions. The authors revised the introduction and showed as following:

  1. Introduction

Regeneration waste rubber has always been a major issue that needs to be solved in many countries around the world [1-3]. The random stacking and disposal of waste rubber will cause serious environmental pollution and wasting of resources [4]. Waste rubber acts a modifier to enhance the mechanical properties of asphalt, which is a good application method of waste tires [5]. Waste rubber modified asphalt (RMA), a mixture of 20% waste rubber and base asphalt, first appeared in a British patent in 1843[6]. However, the compatibility problem between waste rubber and asphalt not only restricts the engineering application of RMA, but also hinders the recycling of waste rubber. Owing to rich aromatics in aromatic oil, aromatic oil activated waste rubber has secured popularity in asphalt materials due to their environmental, economic, and additive advantages, especially in high-content waste rubber modified asphalt (HRMA)[7]..

There are many kinds of aromatic oil, which can active RMA. For example, waste engine oil, as an additive, modified asphalt showing great potential in reducing viscosity and enhancing compatibility [8]. In other hand, the addition of waste engine oil would damage the rutting and aging resistance of asphalt [9]. So, we can concluded that aromatic oil can increase obviously the compatibility, while damage the rutting. Waste mineral oil could improve the compatibility and flexibility of RMA and have a positive effect on the work-ability of RMA [10-11]. Waste cooking oil can obviously reduce the viscosity of RMA, and improve its compatibility, low temperature performance and anti-aging performance of RMA [12]. Waste cooking oil residue can increase significantly the compatibility of RMA. The improved compatibility of RMA incorporating ground tire rubber pee-swelled with waste cooking oil residue was mainly attributed to the extended release of nature rubber and traces of synthetic rubber and inorganic filler into the asphalt phase [13]. Bio-oil activated rubber is also an effective way to improve the compatibility between waste rubber and asphalt [10]. Bio-oil also reduce the viscosity of RMA and improve the work-ability [14]. Segregation was alleviated due to the microwave pre-treatment of bio-oil on crumb rubber and the pre-treatment of bio-oil on crumb rubber facilitated the swelling and depolymerization of crumb rubber in the melt asphalt [15]. Bio-oil grafting rubber particles can increase the crossover modulus after thermal aging and cohesion against moisture damage of RMA [16]. Pyrolysis bio oil can improve the property through rutting and fatigue parameters of RMA. Moreover, pyrolysis bio oil can also reduce the viscous property and increase the elastic property of the asphalt binder [17]. Bio-oil was grafted onto the rubber surface by heat treatment to create surface activated rubber and bio-oil activated RMA can improve the storage stability of rubber modified asphalt. Bio-oil was successfully absorbed by rubber particles significantly improving the dispersion of rubber in asphalt. Both grafting and per-swelling of rubber particles using bio-oil improved properties of RMA [18]. Bio oil can deoxidize and devulcanizing the waste rubber. Moreover, the bio-oil on waste rubber decreases the glass transition temperature of asphalt system, thereby improve the low-temperature properties [19]. Summary, aromatic oil can reduce the viscosity and improve the storage stability of RMA. The viscosity and molecular interaction of HRMA is stronger than that of RMA and has high waste rubber content. So, the viscosity and molecular interaction challenges in HRMA is particularly challenging compared to RMA. However, the molecular interaction mechanism between aromatic oil and HRMA is not clear. The novelty of this work is that molecular dynamic simulations were used to evaluate the molecular interaction mechanism between aromatic oil and HRMA and experimental methods were used to verify the simulated results.

In this study, aromatic oil was used to active the waste rubber, then modify asphalt. Radial distribution function, diffusion coefficient, free volume, solubility parameter, and shear viscosity were calculated by molecular simulations. Storage stability, micro-morphology and adhesive force were measured by experiments. The molecular interaction mechanism between aromatic oil and HRMA was emphatically analyzed.

Point 2:  Regarding the numerical simulation, the software or language employed for the simulations remains unspecified

Response 2: Thank you for your comments. The software used Materials Studio and the language used software native language. The authors added some content in the revised manuscript and it revised as “In order to improve the storage stability, 7% aromatic oil was added into HRMA and it abbreviated as HRMA-2. The dosage of waste rubber and aromatic oil were acquired according to the authors previous study [21]. The molecular models built using Materials Studio software.”.

Point 3: In Fig.3, an explanation is required to understand why HRMA-1 exhibits a higher and sharper first peak intensity when compared to HRMA-2.

Response 3: Thank you for your reviews. The reason of HRMA-1 exhibits a higher and sharper first peak intensity when compared to HRMA-2 is that the asphaltenes content of HRMA-1 is bigger than that of HRMA-2.

Point 4: The description of "Figure 4(b) illustrates the MSD results of all components of HRMA," conflicts with the actual content of the figure

Response 4: Thank you for your suggestions. This is a good question. The authors revised the illustrates as “Figure 4(b) depicts the MSD results of asphaltenes in HRMA. The MSD of asphaltenes changes obviously with increasing simulation time. Furthermore, the MSD of asphaltenes decreases significantly with the increasing waste rubber content in HRMA system, and the MSD of asphaltenes also decreases significantly after adding aromatic oil.”.

Point 5:In Fig.8, the AFM force curve lacks coherence. To facilitate comprehension, it is necessary to display the complete measurement curve, encompassing both the attractive and repulsive curves

Response 5: Thank you for your comments. The authors revised the Fig.8.

Reviewer 4 Report

Title: Molecular Interaction and Viscosity Reduction Mechanism of High-content Waste Rubber Modified Asphalt

Reviewer comments: 

This manuscript has been the focus on Molecular Interaction and Viscosity Reduction Mechanism of High-content Waste Rubber Modified Asphalt. The study is original however I feel that the paper could be improved. Therefore, could you consider some points below for further improvement.

1.              Abstract: Overall, the manuscript abstract appears to present an interesting study focused on addressing the challenges associated with high-content waste rubber modified asphalt (HRMA) by using aromatic oil as an activator for waste rubber. The research combines molecular simulations and experiments to investigate the molecular interaction and viscosity reduction mechanism of HRMA. However, there are a few areas where improvements can be made:

a)   The abstract could be more concise by rephrasing some sentences to reduce wordiness and improve clarity.

b)  Adding quantitative results or numerical values would provide a clearer sense of the study's outcomes.

c)   Explain why HRMA is important and how this study contributes to the existing body of knowledge to strengthen the abstract's impact.

d)  The first sentence can be rephrased for better clarity. Instead of exists with a big viscosity," use has a high viscosity." Similarly, "bad storage stability" can be replaced with "poor storage stability

2.              Introduction: The introduction can be further improved to engage readers and provide a clear context for the research. However, please consider some improvement at the points below:

a)   The introduction contains a vast amount of information on various aromatic oils and their effects on rubber-modified asphalt, which might make the section seem somewhat dense and disorganised. Consider structuring the information more clearly to improve readability.

b)  While the introduction mentions the specific focus on HRMA, it could emphasise more explicitly why addressing the viscosity and molecular interaction challenges in HRMA is particularly crucial or challenging compared to other types of RMA.

c)   The introduction includes a significant number of references, which could be limited to the most relevant and recent ones to avoid overwhelming readers.

d)  Some information is repeated or could be consolidated to make the introduction more concise.

3.              Material & Method: Overall, the material and methodology section provides a sound foundation for the study, outlining the materials used and the methods employed for both molecular simulations and experimental verification. Please consider the following points to enhance the robustness and informativeness of the material and methodology sections:

a)   The steps involved in preparing HRMA samples are slightly difficult to follow due to some unclear phrasing. Consider revising and providing a clearer step-by-step explanation of the preparation process for better understanding.

b)  While the chosen experimental verification methods (storage stability test, adhesive force test, and functional group test) are appropriate, it would be helpful to provide a brief justification for why each method was selected to assess the specific properties of HRMA.

c)   While the material and methodology section is comprehensive in describing the methods used, it lacks an explanation of the expected outcomes or how the data obtained will be analysed to address the research objectives.

4.              Results and Discussion: Please consider the following suggestions to improve this section.

a)   Relate the findings to previous studies mentioned earlier in the manuscript. Discuss similarities and differences in results, and explain how the current study contributes to the existing body of knowledge on HRMA.

b)  The discussion on the viscosity reduction mechanism of HRMA is insightful. However, consider providing a more comprehensive explanation of how each factor (e.g., waste rubber content, aromatic oil, microstructure, and surface roughness) affects the viscosity of HRMA. 

c)   Connect FT-IR results with viscosity reduction. Explain how the observed absorption peaks are related to the changes in HRMA viscosity and how they contribute to the overall understanding of the system.

5.          Conclusion:

a)   Summarise key findings concisely. Don't repeat information that has already been discussed in the results and discussion sections. Consider rephrasing and condensing the conclusions to make them more focused.

b)  Highlight the practical implications that relate to the research objectives, limitations, and future directions or explorations.

6.          Similarity: 29%

a)   Please reduce the similarity of this manuscript.

7.              Language:

a)   Throughout the manuscript, there are some grammatical errors, awkward phrasings, and run-on sentences that need to be addressed. For example, in the sentence "The reason is that aromatic oil wraps the rubber particles and appears in the sol state, which is very similar to the state of asphalt and is easy to move in asphalt," consider rephrasing it to "The reason is that aromatic oil envelops the rubber particles, creating a sol state similar to that of asphalt, which promotes movement within the asphalt."

b)  There are several minor typographical errors and missing spaces between words that should be corrected.

a)   Throughout the manuscript, there are some grammatical errors, awkward phrasings, and run-on sentences that need to be addressed. For example, in the sentence "The reason is that aromatic oil wraps the rubber particles and appears in the sol state, which is very similar to the state of asphalt and is easy to move in asphalt," consider rephrasing it to "The reason is that aromatic oil envelops the rubber particles, creating a sol state similar to that of asphalt, which promotes movement within the asphalt."

b)  There are several minor typographical errors and missing spaces between words that should be corrected.

Author Response

This manuscript has been the focus on Molecular Interaction and Viscosity Reduction Mechanism of High-content Waste Rubber Modified Asphalt. The study is original however I feel that the paper could be improved. Therefore, could you consider some points below for further improvement.

Response: Thank you for your kind suggestions. The authors revised their manuscript according to comments with point-by-point. All changes made to the revised manuscript are in bold and color font. Below you will find our point-by-point responses to the reviewer’s comments/questions.

1.Abstract: Overall, the manuscript abstract appears to present an interesting study focused on addressing the challenges associated with high-content waste rubber modified asphalt (HRMA) by using aromatic oil as an activator for waste rubber. The research combines molecular simulations and experiments to investigate the molecular interaction and viscosity reduction mechanism of HRMA. However, there are a few areas where improvements can be made:

  1. a) The abstract could be more concise by rephrasing some sentences to reduce wordiness andimprove clarity.

Response: Thank you for your reviews. The authors deleted some words or sentences reduce wordiness and improve clarity. The detailed revision showed as following:

High-content waste rubber modified asphalt (HRMA) has big viscosity and poor storage stability. HRMA can not only improve road propereties of asphalt, but also remit the environmental pollution caused by waste tires. How to enhance the molecular interaction of waste rubber and asphalt is key of take full use of HRMA. In this paper, aromatic oil was acted as activator for waste rubber. Molecular interaction mechanism between aromatic oil and HRMA were investigated.

  1. b) Adding quantitative results or numerical values would provide a clearer sense of the study's

Response: The authors added some content about quantitative results as “Storage stability, micro-morphology and adhesive force were measured by experiments. The adhesive force of HRMA-1 (4.9nN) is less than that of RMA (6.2nN) and HRMA-2 (5.8nN). ”.

  1. c) Explain why HRMA is important and how this study contributes to the existing body ofknowledge to strengthen the abstract's impact.

Response: The authors added one sentence to explain why HRMA is important as HRMA can not only improve road propereties of asphalt, but also remit the environmental pollution caused by waste tires. .

  1. d) The first sentence can be rephrased for better clarity. Instead of exists with a big viscosity,"use has a high viscosity." Similarly, "bad storage stability" can be replaced with "poorstorage stability.

Response: Thank you for your suggestions. The authors revised the first sentence as “High-content waste rubber modified asphalt (HRMA) has big viscosity and poor storage stability.”.

2.Introduction: The introduction can be further improved to engage readers and provide a clear context for the research. However, please consider some improvement at the points below:

  1. a) The introduction contains a vast amount of information on various aromatic oils and theireffects on rubber-modified asphalt, which might make the section seem somewhat denseand disorganised. Consider structuring the information more clearly to improve readability.

Response: There are many kinds of aromatic oil, which can active RMA. For example, waste engine oil, as an additive, modified asphalt showing great potential in reducing viscosity and enhancing compatibility [8]. In other hand, the addition of waste engine oil would damage the rutting and aging resistance of asphalt [9]. So, we can concluded that aromatic oil can increase obviously the compatibility, while damage the rutting. Waste mineral oil could improve the compatibility and flexibility of RMA and have a positive effect on the work-ability of RMA [10-11]. Waste cooking oil can obviously reduce the viscosity of RMA, and improve its compatibility, low temperature performance and anti-aging performance of RMA [12]. Waste cooking oil residue can increase significantly the compatibility of RMA. The improved compatibility of RMA incorporating ground tire rubber pee-swelled with waste cooking oil residue was mainly attributed to the extended release of nature rubber and traces of synthetic rubber and inorganic filler into the asphalt phase [13]. Bio-oil activated rubber is also an effective way to improve the compatibility between waste rubber and asphalt [10]. Bio-oil also reduce the viscosity of RMA and improve the work-ability [14]. Segregation was alleviated due to the microwave pre-treatment of bio-oil on crumb rubber and the pre-treatment of bio-oil on crumb rubber facilitated the swelling and depolymerization of crumb rubber in the melt asphalt [15]. Bio-oil grafting rubber particles can increase the crossover modulus after thermal aging and cohesion against moisture damage of RMA [16]. Pyrolysis bio oil can improve the property through rutting and fatigue parameters of RMA. Moreover, pyrolysis bio oil can also reduce the viscous property and increase the elastic property of the asphalt binder [17]. Bio-oil was grafted onto the rubber surface by heat treatment to create surface activated rubber and bio-oil activated RMA can improve the storage stability of rubber modified asphalt. Bio-oil was successfully absorbed by rubber particles significantly improving the dispersion of rubber in asphalt. Both grafting and per-swelling of rubber particles using bio-oil improved properties of RMA [18]. Bio oil can deoxidize and devulcanizing the waste rubber. Moreover, the bio-oil on waste rubber decreases the glass transition temperature of asphalt system, thereby improve the low-temperature properties [19].

  1. b) While the introduction mentions the specific focus on HRMA, it could emphasise moreexplicitly why addressing the viscosity and molecular interaction challenges in HRMA isparticularly crucial or challenging compared to other types of RMA.

Response: The authors added one sentence to explain the reason as The viscosity and molecular interaction of HRMA is stronger than that of RMA and has high waste rubber content. So, the viscosity and molecular interaction challenges in HRMA is particularly challenging compared to RMA. .

  1. c) The introduction includes a significant number of references, which could be limited to themost relevant and recent ones to avoid overwhelming readers.

Response: The references are published recent years! The published years ranged from 2017 to 2023(Except for a classic reference which published at 2014) and they are closely related to the thesis topic. The detailed references showed as following:

References

[1] Modupe Emmanuel Abayomi, Atoyebi Dotun Olumoyewa, Oluwatuyi Emmanuel Opeyemi, Aladegboye James Oluwasegun, Busari Adebola Ayobami, Basorun Ofonime Adebayo. Dataset of mechanical, marshall and rheological properties of crumb rubber-Bio-oil modified hot mix asphalt for sustainable pavement works[J]. Data in Brief, 2018,21:63-70.

[2] Cao Liping, Su Zhibin, Liu Ruirui, Zhou Tao. Optimized formulation of asphalt compound containing bio-oil and shredded rubber[J]. Journal of Cleaner Production, 2022,378:134591.

[3] Qiu Yangke, Gao Yang, Wei Yachun, Cao Junsheng, Wang Xiaolong, Wang Shuhui. Conventional properties, rheological properties, and storage stability of crumb rubber modified asphalt with WCO and ABS[J]. Construction and Building Materials, 2023,392:131987.

[4] Duan Kaixi, Wang Chaohui, Liu Jikang, Song Liang, Chen Qian, Chen Yuanzhao. Research progress and performance evaluation of crumb-rubber-modified asphalts and their mixtures[J]. Construction and Building Materials, 2022,361:129687.

[5] Dong Mengzhen, Dong Ruikun. Reaction mechanism and rheological properties of waste cooking oil pre-desulfurized crumb tire rubber/SBS composite modified asphalt[J]. Construction and Building Materials, 2021,274:122083.

[6] Wu Wangjie, Jiang Wei, Xiao Jingjing, Yuan Dongdong, Wang Teng, Xing Chengwei. Analysis of thermal susceptibility and rheological properties of asphalt binder modified with microwave activated crumb rubber[J]. Journal of Cleaner Production, 2022,377:134488.

[7] Yang Huifang, Dong Ruikun. Investigating the properties of rejuvenated asphalt with the modified rejuvenator prepared by waste cooking oil and waste tire crumb rubber[J]. Construction and Building Materials, 2022,315:125692.

[8] Wang Shifeng, Chen Dingxin, Xiao Feipeng. Recent developments in the application of chemical approaches to rubberized asphalt[J]. Construction and Building Materials, 2017,131:101-113.

[9] Liu Qi, Han Bo, Wang Shuyi, Falchetto Cannone Augusto, Wang Di, Yu Bin, Zhang Jiupeng. Evaluation and molecular interaction of asphalt modified by rubber particles and used engine oil[J]. Journal of Cleaner Production, 2022,375:134222.

[10] Faisal Kabirb Sk, Zakertabrizi Mohammad, Hosseini Ehsan, Fini Elham. Effects of Amide-Based modifiers on surface activation and devulcanization of rubber[J]. Computational Materials Science, 2021,188:110175.

[11] Yan Yong, Guo Rongxin, Liu Zhuo, Yang Yang, Wang Xiaoyong. Property improvement of thermosetting natural rubber asphalt binder by mineral oil[J]. Journal of Materials Research and Technology, 2023,24:8807-8825.

[12] Feng Xinjun, Liang Hui, Dai Zijian. Rheological properties and microscopic mechanism of waste cooking oil activated waste crumb rubber modified asphalt[J]. Journal of Road Engineering, 2022,2:357-368.

[13] Ma Jianmin, Hu Mingjun, Sun Daquan, Lu Tong, Sun Guoqiang, Ling Senlin, Xu Lei. Understanding the role of waste cooking oil residue during the preparation of rubber asphalt[J]. Resources, Conservation & Recycling, 2021,167:105235.

[14] Wang Hui, Jing Yufei, Zhang Jiupeng, Cao Yuanbo, Lyu Lei. Preparation and performance evaluation of swine manure bio-oil modified rubber asphalt binder[J]. Construction and Building Materials, 2021,294:123584.

[15] Lei Yong, Wei Zonghuaxuan, Wang Hainian, You Zhanping, Yang Xu, Chen Yu. Effect of crumb rubber size on the performance of rubberized asphalt with bio-oil pretreatment[J]. Construction and Building Materials, 2021,285:122864.

[16] Lyu Lei, Pei Jianzhong, Hu Dongliang, Sun Guoqing, Fini Elham. Bio-modified rubberized asphalt binder: A clean, sustainable approach to recycle rubber into construction[J]. Journal of Cleaner Production, 2022,345:131151.

[17] Kumar Ankush, Choudhary Rajan, Kumar Abhinay. Composite asphalt binder modification with waste Non-tire automotive rubber and pyrolytic oil[J]. Materials Today: Proceedings, 2022,61:158-166.

[18] Zhou Tao, Faisal Kabirb Sk, Cao Liping, Luan Hai, Dong Zejiao, Fini Elham. Comparing effects of physisorption and chemisorption of bio-oil onto rubber particles in asphalt[J]. Journal of Cleaner Production, 2020,273:123112.

[19] Lyu Lei, Mikhailenko Peter, Piao Zhengyin, Fini Elham, Pei Jianzhong, Poulikakos Lily. Unraveling the modifcation mechanisms of waste bio-oils and crumb rubber on asphalt binder based on microscopy and chemo-rheology[J]. Resources, Conservation & Recycling, 2022,185:106447.

[20] Li Derek, Greenfield Michael. Chemical compositions of improved model asphalt systems for molecular simulations[J]. Fuel, 2014,115:347-356.

[21] Zhou Xinxing. Physicochemical properties of high-content rubber modified bio-asphalt using molecular simulation[J].Petroleum Science and Technology, 2023,2210601.

[22] Ren Shisong, Liu Xueyan, Gao Yangming, Jin Ruxin, Lin Peng, Erkens Sandra, Wang Haopeng. Molecular dynamics simulation and experimental validation on the interfacial diffusion behaviors of rejuvenators in aged bitumen[J]. Materials & Design, 2023,226:111619.

[23] Ren Shisong, Liu Xueyan, Lin Pang, Erkens Sandra, Gao Yangming. Chemical characterizations and molecular dynamics simulations on different rejuvenators for aged bitumen recycling[J]. Fuel, 2022,324:124550.

[24] Duan Kaixi, Wang Chaohui, Liu Jikang, Song Liang, Chen Qian, Chen Yuanzhao. Research progress and performance evaluation of crumb-rubber-modified asphalts and their mixtures[J]. Construction and Building Materials, 2022,361:129687.

[25] Zhou Xinxing, Zhao Guangyuan, Miljković Miomir, Tighe Susan, Chen Meizhu , Wu Shaopeng. Crystallization kinetics and morphology of biochar modified bio-asphalt binder[J].Journal of Cleaner Production, 2022, 349:131495.

[26] Eltwati Ahmed, AI-Saffar Zaid, Mohamed Azman, Hainin Roslimohd, Elnihum Ahmed, Enieb Mahmoud. Synergistic effect of SBS copolymers and aromatic oil on the characteristics of asphalt binders and mixtures containing reclaimed asphalt pavement[J]. Construction and Building Materials, 2022,327:127026.

  1. d) Some information is repeated or could be consolidated to make the introduction more

Response: Thank you for your comments. The authors deleted the repeated information. The detailed revision showed as following:

Regeneration waste rubber has always been a major issue that needs to be solved in many countries around the world [1-3]. The random stacking and disposal of waste rubber will cause serious environmental pollution and wasting of resources [4]. Waste rubber acts a modifier to enhance the mechanical properties of asphalt, which is a good application method of waste tires [5]. Waste rubber modified asphalt (RMA), a mixture of 20% waste rubber and base asphalt, first appeared in a British patent in 1843[6]. However, the compatibility problem between waste rubber and asphalt not only restricts the engineering application of RMA, but also hinders the recycling of waste rubber. Owing to rich aromatics in aromatic oil, aromatic oil activated waste rubber has secured popularity in asphalt materials due to their environmental, economic, and additive advantages, especially in high-content waste rubber modified asphalt (HRMA)[7].

There are many kinds of aromatic oil, which can active RMA. For example, waste engine oil, as an additive, modified asphalt showing great potential in reducing viscosity and enhancing compatibility [8]. In other hand, the addition of waste engine oil would damage the rutting and aging resistance of asphalt [9]. So, we can concluded that aromatic oil can increase obviously the compatibility, while damage the rutting. Waste mineral oil could improve the compatibility and flexibility of RMA and have a positive effect on the work-ability of RMA [10-11]. Waste cooking oil can obviously reduce the viscosity of RMA, and improve its compatibility, low temperature performance and anti-aging performance of RMA [12]. Waste cooking oil residue can increase significantly the compatibility of RMA. The improved compatibility of RMA incorporating ground tire rubber pee-swelled with waste cooking oil residue was mainly attributed to the extended release of nature rubber and traces of synthetic rubber and inorganic filler into the asphalt phase [13]. Bio-oil activated rubber is also an effective way to improve the compatibility between waste rubber and asphalt [10]. Bio-oil also reduce the viscosity of RMA and improve the work-ability [14]. Segregation was alleviated due to the microwave pre-treatment of bio-oil on crumb rubber and the pre-treatment of bio-oil on crumb rubber facilitated the swelling and depolymerization of crumb rubber in the melt asphalt [15]. Bio-oil grafting rubber particles can increase the crossover modulus after thermal aging and cohesion against moisture damage of RMA [16]. Pyrolysis bio oil can improve the property through rutting and fatigue parameters of RMA. Moreover, pyrolysis bio oil can also reduce the viscous property and increase the elastic property of the asphalt binder [17]. Bio-oil was grafted onto the rubber surface by heat treatment to create surface activated rubber and bio-oil activated RMA can improve the storage stability of rubber modified asphalt. Bio-oil was successfully absorbed by rubber particles significantly improving the dispersion of rubber in asphalt. Both grafting and per-swelling of rubber particles using bio-oil improved properties of RMA [18]. Bio oil can deoxidize and devulcanizing the waste rubber. Moreover, the bio-oil on waste rubber decreases the glass transition temperature of asphalt system, thereby improve the low-temperature properties [19]. Summary, aromatic oil can reduce the viscosity and improve the storage stability of RMA. The viscosity and molecular interaction of HRMA is stronger than that of RMA and has high waste rubber content. So, the viscosity and molecular interaction challenges in HRMA is particularly challenging compared to RMA. However, the molecular interaction mechanism between aromatic oil and HRMA is not clear. The novelty of this work is that molecular dynamic simulations were used to evaluate the molecular interaction mechanism between aromatic oil and HRMA and experimental methods were used to verify the simulated results.

In this study, aromatic oil was used to active the waste rubber, then modify asphalt. Radial distribution function, diffusion coefficient, free volume, solubility parameter, and shear viscosity were calculated by molecular simulations. Storage stability, micro-morphology and adhesive force were measured by experiments. The molecular interaction mechanism between aromatic oil and HRMA was emphatically analyzed.

3.Material & Method: Overall, the material and methodology section provides a sound foundation for the study, outlining the materials used and the methods employed for both molecular simulations and experimental verification. Please consider the following points to enhance the robustness and informativeness of the material and methodology sections:

  1. a) The steps involved in preparing HRMA samples are slightly difficult to follow due to someunclear phrasing. Consider revising and providing a clearer step-by-step explanation of thepreparation process for better understanding.

Response:  Heat 70# base asphalt to flow state at the temperature 130°C~135°C and pour it into the stainless steel sample holder. Add aromatic oil accounting for 20% or 35% of waste rubber with microwave irradiation at 1000W (irradiation power) and 60s (irradiation time). Then, the aromatic oil activated waste rubber was added into asphalt and heat to 170°C~180°C. Finally, a high speed shear meter was used for continuous shearing for 60min at a rate of 2000r/min and a temperature of 170°C~180°C to get the HRMA sample.

  1. b) While the chosen experimental verification methods (storage stability test, adhesive forcetest, and functional group test) are appropriate, it would be helpful to provide a briefjustification for why each method was selected to assess the specific properties of HRMA.

Response: Thank you for your comments. The authors added some content about the verification methods and it showed as following:

Storage stability test of HRMA was evaluated by segregation experiment and micro-morphology. The segregation experiment of HRMA were conducted according to JTG E40-2014.

The adhesive force of HRMA was measured by atomic force microscopy (AFM). The sample size was selected as 10μm×10μm, the tip radius of the probe was 8nm, and the tip height was 10μm~15μm. In the process of testing the tip of the probe, the phenomenon of jumping contact and jumping off will form force curve, and the adhesive force can be obtained by subtracting the lowest force from the highest point of the force curve. The test of adhesive force test according to previous study[19].

  1. c) While the material and methodology section is comprehensive in describing the methodsused, it lacks an explanation of the expected outcomes or how the data obtained will beanalysed to address the research objectives.

Response: Thank you for your reviews. The expected outcomes of data from simulation section can be put into the materials and experiments section. While the expected outcomes of data from experiments can be put into the results and discussions..

4.Results and Discussion: Please consider the following suggestions to improve this section.

  1. a) Relate the findings to previous studies mentioned earlier in the manuscript. Discuss similarities and differences in results, and explain how the current study contributes to the existing body of knowledge on HRMA.

Response: The authors added some content in the results and discussions section and it showed as following:

The results indicated that aromatic oil could change the orderliness and arrangement of molecules structure in HRMA and be contributed to dissolve rubber into asphalt. The reason is that aromatic oil envelops the rubber particles, creating a sol state similar to that of asphalt,which promotes movement within the asphalt. The conclusion is also consistent with the previous study[15].

Previous study thought that the four components of RMA had different first peak positions before and after adding waste rubber or aromatic oil [15]. The first peak position of asphaltenes in RMA (20% waste rubber content), HRMA-1 (35% waste rubber content), and HRMA-2 (35% waste rubber content with 7% aromatic oil) appeared at 0.61Å, 0.59Å, and 0.62Å, respectively.

The intensity of the first peak represented the packing density of asphalt molecules and was related to the number of molecules [22]. The higher and sharper first peak intensity indicated a more structure and regular orderliness of four components molecules, which showed that the addition of waste rubber or aromatic oil made big difference in the aggregation and orderliness of asphalt.

The results demonstrate that waste rubber can inhibit the asphalt molecules movement, while the aromatic oil will promote the asphalt molecules movement and accelerate the dispersion of waste rubber in asphalt system. The conclusion is also consistent with the previous study[21].

Furthermore, both rubber content and aromatic oil would affect significantly the FV of HRMA. Usually, the more the size of the Connolly radius is, the less the FV of the HRMA system [23]. So, the Connolly radius of HRMA would change with the increase of rubber content or the addition of aromatic oil. The FV of HRMA is above 5%, which indicates that there is a critical FV to ensure that the molecules of HRMA system can move freely.

The addition of the aromatic oil to HRMA is beneficial to reduce the difference of δ between rubber and asphalt system to improve the compatibility of components. This conclusion is consistent with previous study[22].

Previous research found that asphaltenes content are related to viscosity of asphalt and the more the asphaltenes content is, the bigger the viscosity of asphalt is[24]. The above results is in good agreement with previous conclusions.

The catana-phase increases with the addition of aromatic oil in HRMA. Moreover, the peri-phase is the major structure phase in HRMA. Previous research found that catana-phase was the crystallization wax or asphaltenes [25]. The results show that aromatic oil maybe exist the crystallization wax.

Although the carbonyl functional group was not obvious in the HRMA samples, the sulfoxide absorption peak gradually increased after addition of aromatic oil. These findings are in agreement with previous results[26]. Owing to the viscosity of 7% aromatic oil in HRMA is less than that of HRMA, we can concluded that the sulfoxide absorption peak can reduce the viscosity of HRMA.

  1. b) The discussion on the viscosity reduction mechanism of HRMA is insightful. However,consider providing a more comprehensive explanation of how each factor (e.g., waste rubber content, aromatic oil, microstructure, and surface roughness) affects the viscosity of HRMA.

Response: Thank you, this is a very good review for us. The authors revised and added some content about viscosity reduction mechanism of HRMA and it showed as following:

On the one hand, the relative molecular mass and the volume of waste rubber are also greater than the average values of asphalt molecules, so the rubber molecules make HRMA more crowded and harder to move relatively. On the other hand, the aromatic oil molecules can dissolve some molecules with polarity in asphalt and promote the mobility of the asphalt system.

Moreover, there exists strong the electrostatic force between rubber and asphaltenes and the intermolecular force between rubber and aromatic oil or aromatics, which makes the aromatic oil and aromatics parcel rubber molecules and makes the waste rubber highly soluble in asphalt. It indicates that aromatic oil can promote the dispersion of waste rubber, making the asphalt system stable storage.

The results indicate that the viscosity of yellow transparent phase is less than that of black aggregate. Moreover, the viscosity reduction mechanism of HRMA from the morphology perspective shows that the addition of aromatic oil can increase the yellow transparent phase and decrease the black aggregate to change the morphology of HRMA and reduce the viscosity of HRMA. The FM morphology of HRMA is related to the viscosity.

The results show that aromatic oil maybe exist the crystallization wax. However, the crystallization wax is a kind of low-viscosity aromatic oil. So, the addition of aromatic oil can reduce the viscosity of HRMA by increasing the crystallization wax.

Owing to the viscosity of 7% aromatic oil in HRMA is less than that of HRMA, we can concluded that the reduce of sulfoxide absorption peak and the exist of carbonyl peak can reduce the viscosity of HRMA.

  1. c) Connect FT-IR results with viscosity reduction. Explain how the observed absorption peaks are related to the changes in HRMA viscosity and how they contribute to the overall understanding of the system.

Response: Thank you for your suggestions. The authors revised the content of FT-IR section as “The greatest difference between the HRMA and 7% aromatic oil in HRMA was the band of sulfoxide group (S=O) near 1029cm-1. Moreover, there exists carbonyl (1720cm-1). The sulfoxide absorption peak gradually decreased and carbonyl increased after addition of aromatic oil. These findings are in agreement with previous results[26]. Based on the previous results, the viscosity of 7% aromatic oil in HRMA is less than that of HRMA, we can concluded that the reduce of sulfoxide absorption peak and the exist of carbonyl peak can reduce the viscosity of HRMA.”.

5.Conclusion:

  1. a) Summarise key findings concisely. Don't repeat information that has already been discussed in the results and discussion sections. Consider rephrasing and condensing the conclusions to make them more focused.

Response: The molecular interaction mechanism of HRMA were investigated by molecular dynamic simulations and experiments. We can be drawn the following conclusions:

Aromatic oil can change the orderliness and arrangement of molecules structure in HRMA and be contributed to dissolve rubber into asphalt. Furthermore, the addition of aromatic oil makes big difference in the aggregation and orderliness of asphalt. The addition of aromatic oil can promote the molecular movement and increase the diffusion coefficients.

Aromatic oil can promote the dispersion of waste rubber and asphaltenes, making the HRMA more stable storage. Moreover, molecular simulations confirm the exists molecular interaction between rubber and aromatic oil, and aromatic oil can reduce the viscosity of HRMA.

  1. b) Highlight the practical implications that relate to the research objectives, limitations, and future directions or explorations.

Response: Thank you for your comments. The authors thought that the future directions or explorations showed as following:

  • The interaction mechanism between aromatic oil and rubber modified asphalt was investigated by first-principle.
  • The warm-mixing effect of aromatic oil activated rubber modified asphalt was investigated.
  • The activated effect of aromatic oil on waste rubber was investigated.

6.Similarity: 29%

  1. a) Please reduce the similarity of this manuscript.

Response: Thank you for your comments. The authors had revised the manuscript and in order to reduce the similarity of this manuscript. We hope that the revised manuscript can meet the reviewers’ requirements.

7.Language:

  1. a) Throughout the manuscript, there are some grammatical errors, awkward phrasings, and run-on sentences that need to be addressed. For example, in the sentence "The reason is that aromatic oil wraps the rubber particles and appears in the sol state, which is very similar to the state of asphalt and is easy to move in asphalt," consider rephrasing it to "The reason is that aromatic oil envelops the rubber particles, creating a sol state similar to that of asphalt,which promotes movement within the asphalt."

Response: Thank you for your suggestions. Thank you very much. The authors revised the manuscript according to the reviewer and revised the sentence as “The reason is that aromatic oil envelops the rubber particles, creating a sol state similar to that of asphalt,which promotes movement within the asphalt.”.

  1. b) There are several minor typographical errors and missing spacesbetween words that should be corrected.

Response: Thank you for your attention. The authors deleted the missing spaces and revised the minor typographical errors in the revised manuscript.

Round 2

Reviewer 1 Report

No further comments as all my comments were addressed by the authors 

Author Response

Thank you for your review, all your suggestions are very important to us. 

Reviewer 2 Report

Enclosed are my comments.

Author Response

Point 1: You should mention all limitation(s) you had in this study and suggestion(s) you have for the future studies in the text body.

Response 1: Thank you for your kind suggestions. The authors have realized that the limitations and the suggestions for our future studies should be mentioned in the text body, so we added this part in line 421, 433-440.

Point 2:  The similarity rate is high! This means your work has extensive overlap with other published works and then needs full, accurate reconsideration

Response 2: Thank you for your review. The authors did not submit any similar works before, we have no idea why the similarity rate is so high, but we have revised carefully this manuscript again to reduced the similarity. We hope that the revised manuscript can meet the journal’s requirements.

Reviewer 3 Report

1. Please include the brand and model of the AFM utilized in the experiments.

Significant revisions are necessary to improve clarity and correct grammar errors, e.g. lines 71-72:  the viscosity and molecular interaction challenges in HRMA is particularly challenging compared to RMA.

Author Response

Point 1:  Please include the brand and model of the AFM utilized in the experiments.

Response 1: Thank you for your reviews, we included the brand and model of the AFM in line 191,  which showed as “Brand: Bruker;  Model: Dimension FastScan) ”.

Point 2: Significant revisions are necessary to improve clarity and correct grammar errors, e.g. lines 71-72:  the viscosity and molecular interaction challenges in HRMA is particularly challenging compared to RMA.

Response 2: Thank you for your comments. The authors have revised the  lines 71-72 which showed as “the relationship between viscosity and molecular interaction in HRMA is particularly challenging compared to RMA”. Besides, we have checked and revised the text body again to meet the journal’s requirements.  Thank you again for your kind suggestions.
